# Combinatorial Cascading Bandits

**Branislav Kveton**
Adobe Research
San Jose, CA
*kveton@adobe.com*

**Zheng Wen**
Yahoo Labs
Sunnyvale, CA
*zhengwen@yahoo-inc.com*

**Azin Ashkan**
Technicolor Research
Los Altos, CA
*azin.ashkan@technicolor.com*

**Csaba Szepesvári**
Department of Computing Science
University of Alberta
*szepesva@cs.ualberta.ca*

## Abstract

We propose *combinatorial cascading bandits*, a class of partial monitoring problems where at each step a learning agent chooses a tuple of ground items subject to constraints and receives a reward if and only if the weights of all chosen items are one. The weights of the items are binary, stochastic, and drawn independently of each other. The agent observes the index of the first chosen item whose weight is zero. This observation model arises in network routing, for instance, where the learning agent may only observe the first link in the routing path which is down, and blocks the path. We propose a UCB-like algorithm for solving our problems, `CombCascade`; and prove gap-dependent and gap-free upper bounds on its $n$-step regret. Our proofs build on recent work in stochastic combinatorial semi-bandits but also address two novel challenges of our setting, a non-linear reward function and partial observability. We evaluate `CombCascade` on two real-world problems and show that it performs well even when our modeling assumptions are violated. We also demonstrate that our setting requires a new learning algorithm.

## 1   Introduction

Combinatorial optimization [16] has many real-world applications. In this work, we study a class of combinatorial optimization problems with a binary objective function that returns one if and only if the weights of all chosen items are one. The weights of the items are binary, stochastic, and drawn independently of each other. Many popular optimization problems can be formulated in our setting. Network routing is a problem of choosing a routing path in a computer network that maximizes the probability that all links in the chosen path are up. Recommendation is a problem of choosing a list of items that minimizes the probability that none of the recommended items are attractive. Both of these problems are closely related and can be solved using similar techniques (Section 2.3).

*Combinatorial cascading bandits* are a novel framework for online learning of the aforementioned problems where the distribution over the weights of items is unknown. Our goal is to maximize the expected cumulative reward of a learning agent in $n$ steps. Our learning problem is challenging for two main reasons. First, the reward function is non-linear in the weights of chosen items. Second, we only observe the index of the first chosen item with a zero weight. This kind of feedback arises frequently in network routing, for instance, where the learning agent may only observe the first link in the routing path which is down, and blocks the path. This feedback model was recently proposed in the so-called *cascading bandits* [10]. The main difference in our work is that the feasible set can be arbitrary. The feasible set in cascading bandits is a uniform matroid.

Stochastic online learning with combinatorial actions has been previously studied with semi-bandit feedback and a linear reward function [8, 11, 12], and its monotone transformation [5]. Established algorithms for multi-armed bandits, such as UCB1 [3], KL-UCB [9], and Thompson sampling [18, 2]; can be usually easily adapted to stochastic combinatorial semi-bandits. However, it is non-trivial to show that the algorithms are statistically efficient, in the sense that their regret matches some lower bound. Kveton *et al.* [12] recently showed this for CombUCB1, a form of UCB1. Our analysis builds on this recent advance but also addresses two novel challenges of our problem, a non-linear reward function and partial observability. These challenges cannot be addressed straightforwardly based on Kveton *et al.* [12, 10].

We make multiple contributions. In Section 2, we define the online learning problem of *combinatorial cascading bandits* and propose CombCascade, a variant of UCB1, for solving it. CombCascade is computationally efficient on any feasible set where a linear function can be optimized efficiently. A minor-looking improvement to the UCB1 upper confidence bound, which exploits the fact that the expected weights of items are bounded by one, is necessary in our analysis. In Section 3, we derive gap-dependent and gap-free upper bounds on the regret of CombCascade, and discuss the tightness of these bounds. In Section 4, we evaluate CombCascade on two practical problems and show that the algorithm performs well even when our modeling assumptions are violated. We also show that CombUCB1 [8, 12] cannot solve some instances of our problem, which highlights the need for a new learning algorithm.

## 2 Combinatorial Cascading Bandits

This section introduces our learning problem, its applications, and also our proposed algorithm. We discuss the computational complexity of the algorithm and then introduce the co-called *disjunctive* variant of our problem. We denote random variables by boldface letters. The cardinality of set $A$ is $|A|$ and we assume that $\min \emptyset = +\infty$. The binary and operation is denoted by $\wedge$, and the binary or is $\vee$.

### 2.1 Setting

We model our online learning problem as a combinatorial cascading bandit. A *combinatorial cascading bandit* is a tuple $B = (E, P, \Theta)$, where $E = \{1, \ldots, L\}$ is a finite set of $L$ *ground items*, $P$ is a probability distribution over a binary hypercube $\{0, 1\}^E$, $\Theta \subseteq \Pi^*(E)$, and:

$$\Pi^*(E) = \{(a_1, \ldots, a_k) : k \geq 1, \ a_1, \ldots, a_k \in E, \ a_i \neq a_j \text{ for any } i \neq j\}$$

is the set of all tuples of distinct items from $E$. We refer to $\Theta$ as the *feasible set* and to $A \in \Theta$ as a *feasible solution*. We abuse our notation and also treat $A$ as the *set of items* in solution $A$. Without loss of generality, we assume that the feasible set $\Theta$ covers the ground set, $E = \cup \Theta$.

Let $(\mathbf{w}_t)_{t=1}^n$ be an i.i.d. sequence of $n$ *weights* drawn from distribution $P$, where $\mathbf{w}_t \in \{0, 1\}^E$. At time $t$, the learning agent chooses solution $\mathbf{A}_t = (\mathbf{a}_1^t, \ldots, \mathbf{a}_{|\mathbf{A}_t|}^t) \in \Theta$ based on its past observations and then receives a *binary reward*:

$$\mathbf{r}_t = \min_{e \in \mathbf{A}_t} \mathbf{w}_t(e) = \bigwedge_{e \in \mathbf{A}_t} \mathbf{w}_t(e)$$

as a response to this choice. The reward is one if and only if the weights of *all* items in $\mathbf{A}_t$ are one. The key step in our solution and its analysis is that the reward can be expressed as $\mathbf{r}_t = f(\mathbf{A}_t, \mathbf{w}_t)$, where $f : \Theta \times [0, 1]^E \to [0, 1]$ is a *reward function*, which is defined as:

$$f(A, w) = \prod_{e \in A} w(e), \quad A \in \Theta, \quad w \in [0, 1]^E.$$

At the end of time $t$, the agent observes the index of the first item in $\mathbf{A}_t$ whose weight is zero, and $+\infty$ if such an item does not exist. We denote this feedback by $\mathbf{O}_t$ and define it as:

$$\mathbf{O}_t = \min \left\{ 1 \leq k \leq |\mathbf{A}_t| : \mathbf{w}_t(\mathbf{a}_k^t) = 0 \right\}.$$

Note that $\mathbf{O}_t$ fully determines the weights of the first $\min \{\mathbf{O}_t, |\mathbf{A}_t|\}$ items in $\mathbf{A}_t$. In particular:

$$\mathbf{w}_t(\mathbf{a}_k^t) = \mathbb{1}\{k < \mathbf{O}_t\} \quad k = 1, \ldots, \min \{\mathbf{O}_t, |\mathbf{A}_t|\} . \tag{1}$$

Accordingly, we say that item $e$ is *observed* at time $t$ if $e = \mathbf{a}_k^t$ for some $1 \le k \le \min\{\mathbf{O}_t, |\mathbf{A}_t|\}$. Note that the order of items in $\mathbf{A}_t$ affects the feedback $\mathbf{O}_t$ but not the reward $\mathbf{r}_t$. This differentiates our problem from combinatorial semi-bandits.

The goal of our learning agent is to maximize its expected cumulative reward. This is equivalent to minimizing the *expected cumulative regret* in $n$ steps:

$$R(n) = \mathbb{E}\left[\sum_{t=1}^n R(\mathbf{A}_t, \mathbf{w}_t)\right],$$

where $R(\mathbf{A}_t, \mathbf{w}_t) = f(A^*, \mathbf{w}_t) - f(\mathbf{A}_t, \mathbf{w}_t)$ is the *instantaneous stochastic regret* of the agent at time $t$ and $A^* = \arg\max_{A \in \Theta} \mathbb{E}[f(A, \mathbf{w})]$ is the *optimal solution* in hindsight of knowing $P$. For simplicity of exposition, we assume that $A^*$, as a set, is unique.

A major simplifying assumption, which simplifies our optimization problem and its learning, is that the distribution $P$ is factored:

$$P(w) = \prod_{e \in E} P_e(w(e)), \tag{2}$$

where $P_e$ is a Bernoulli distribution with mean $\bar{w}(e)$. We borrow this assumption from the work of Kveton *et al.* [10] and it is critical to our results. We would face computational difficulties without it. Under this assumption, the *expected reward* of solution $A \in \Theta$, the probability that the weight of each item in $A$ is one, can be written as $\mathbb{E}[f(A, \mathbf{w})] = f(A, \bar{w})$, and depends only on the expected weights of individual items in $A$. It follows that:

$$A^* = \arg\max_{A \in \Theta} f(A, \bar{w}).$$

In Section 4, we experiment with two problems that violate our independence assumption. We also discuss implications of this violation.

Several interesting online learning problems can be formulated as combinatorial cascading bandits. Consider the problem of learning routing paths in *Simple Mail Transfer Protocol (SMTP)* that maximize the probability of e-mail delivery. The ground set in this problem are all links in the network and the feasible set are all routing paths. At time $t$, the learning agent chooses routing path $\mathbf{A}_t$ and observes if the e-mail is delivered. If the e-mail is not delivered, the agent observes the first link in the routing path which is down. This kind of information is available in SMTP. The weight of item $e$ at time $t$ is an indicator of link $e$ being up at time $t$. The independence assumption in (2) requires that all links fail independently. This assumption is common in the existing network routing models [6]. We return to the problem of network routing in Section 4.2.

## 2.2 `CombCascade` **Algorithm**

Our proposed algorithm, `CombCascade`, is described in Algorithm 1. This algorithm belongs to the family of UCB algorithms. At time $t$, `CombCascade` operates in three stages. First, it computes the *upper confidence bounds (UCBs)* $\mathbf{U}_t \in [0,1]^E$ on the expected weights of all items in $E$. The UCB of item $e$ at time $t$ is defined as:

$$\mathbf{U}_t(e) = \min\left\{\hat{\mathbf{w}}_{\mathbf{T}_{t-1}(e)}(e) + c_{t-1, \mathbf{T}_{t-1}(e)}, 1\right\}, \tag{3}$$

where $\hat{\mathbf{w}}_s(e)$ is the average of $s$ observed weights of item $e$, $\mathbf{T}_t(e)$ is the number of times that item $e$ is observed in $t$ steps, and $c_{t,s} = \sqrt{(1.5 \log t)/s}$ is the radius of a confidence interval around $\hat{\mathbf{w}}_s(e)$ after $t$ steps such that $\bar{w}(e) \in [\hat{\mathbf{w}}_s(e) - c_{t,s}, \hat{\mathbf{w}}_s(e) + c_{t,s}]$ holds with a high probability. After the UCBs are computed, `CombCascade` chooses the optimal solution with respect to these UCBs:

$$\mathbf{A}_t = \arg\max_{A \in \Theta} f(A, \mathbf{U}_t).$$

Finally, `CombCascade` observes $\mathbf{O}_t$ and updates its estimates of the expected weights based on the weights of the observed items in (1), for all items $\mathbf{a}_k^t$ such that $k \le \mathbf{O}_t$.

For simplicity of exposition, we assume that `CombCascade` is initialized by one sample $\mathbf{w}_0 \sim P$. If $\mathbf{w}_0$ is unavailable, we can formulate the problem of obtaining $\mathbf{w}_0$ as an optimization problem on $\Theta$ with a linear objective [12]. The initialization procedure of Kveton *et al.* [12] tracks observed items and adaptively chooses solutions with the maximum number of unobserved items. This approach is computationally efficient on any feasible set $\Theta$ where a linear function can be optimized efficiently.

`CombCascade` has two attractive properties. First, the algorithm is *computationally efficient*, in the sense that $\mathbf{A}_t = \arg\max_{A \in \Theta} \sum_{e \in A} \log(\mathbf{U}_t(e))$ is the problem of maximizing a linear function on

---
**Algorithm 1** `CombCascade` for combinatorial cascading bandits.
---
// Initialization
Observe $\mathbf{w}_0 \sim P$
$\forall e \in E : \mathbf{T}_0(e) \leftarrow 1$
$\forall e \in E : \hat{\mathbf{w}}_1(e) \leftarrow \mathbf{w}_0(e)$

**for all** $t = 1, \ldots, n$ **do**
    // Compute UCBs
    $\forall e \in E : \mathbf{U}_t(e) = \min \left\{ \hat{\mathbf{w}}_{\mathbf{T}_{t-1}(e)}(e) + c_{t-1, \mathbf{T}_{t-1}(e)}, 1 \right\}$

    // Solve the optimization problem and get feedback
    $\mathbf{A}_t \leftarrow \arg\max_{A \in \Theta} f(A, \mathbf{U}_t)$
    Observe $\mathbf{O}_t \in \{1, \ldots, |\mathbf{A}_t|, +\infty\}$

    // Update statistics
    $\forall e \in E : \mathbf{T}_t(e) \leftarrow \mathbf{T}_{t-1}(e)$
    **for all** $k = 1, \ldots, \min \{\mathbf{O}_t, |\mathbf{A}_t|\}$ **do**
        $e \leftarrow \mathbf{a}_k^t$
        $\mathbf{T}_t(e) \leftarrow \mathbf{T}_t(e) + 1$
        $\hat{\mathbf{w}}_{\mathbf{T}_t(e)}(e) \leftarrow \dfrac{\mathbf{T}_{t-1}(e)\hat{\mathbf{w}}_{\mathbf{T}_{t-1}(e)}(e) + \mathbb{1}\{k < \mathbf{O}_t\}}{\mathbf{T}_t(e)}$
---

$\Theta$. This problem can be solved efficiently for various feasible sets $\Theta$, such as matroids, matchings, and paths. Second, `CombCascade` is *sample efficient* because the UCB of solution $A$, $f(A, \mathbf{U}_t)$, is a product of the UCBs of all items in $A$, which are estimated separately. The regret of `CombCascade` does not depend on $|\Theta|$ and is polynomial in all other quantities of interest.

### 2.3 Disjunctive Objective

Our reward model is *conjuctive*, the reward is one if and only if the weights of all chosen items are one. A natural alternative is a *disjunctive* model $\mathbf{r}_t = \max_{e \in \mathbf{A}_t} \mathbf{w}_t(e) = \bigvee_{e \in \mathbf{A}_t} \mathbf{w}_t(e)$, the reward is one if the weight of *any* item in $\mathbf{A}_t$ is one. This model arises in recommender systems, where the recommender is rewarded when the user is satisfied with *any* recommended item. The feedback $\mathbf{O}_t$ is the index of the first item in $\mathbf{A}_t$ whose weight is one, as in cascading bandits [10].

Let $f_\vee : \Theta \times [0,1]^E \to [0,1]$ be a reward function, which is defined as $f_\vee(A, w) = 1 - \prod_{e \in A}(1 - w(e))$. Then under the independence assumption in (2), $\mathbb{E}[f_\vee(A, \mathbf{w})] = f_\vee(A, \bar{w})$ and:

$$A^* = \arg\max_{A \in \Theta} f_\vee(A, \bar{w}) = \arg\min_{A \in \Theta} \prod_{e \in A}(1 - \bar{w}(e)) = \arg\min_{A \in \Theta} f(A, 1 - \bar{w}).$$

Therefore, $A^*$ can be learned by a variant of `CombCascade` where the observations are $1 - \mathbf{w}_t$ and each UCB $\mathbf{U}_t(e)$ is substituted with a *lower confidence bound (LCB)* on $1 - \bar{w}(e)$:

$$\mathbf{L}_t(e) = \max \left\{ 1 - \hat{\mathbf{w}}_{\mathbf{T}_{t-1}(e)}(e) - c_{t-1, \mathbf{T}_{t-1}(e)}, 0 \right\}.$$

Let $R(\mathbf{A}_t, \mathbf{w}_t) = f(\mathbf{A}_t, 1 - \mathbf{w}_t) - f(A^*, 1 - \mathbf{w}_t)$ be the instantaneous stochastic regret at time $t$. Then we can bound the regret of `CombCascade` as in Theorems 1 and 2. The only difference is that $\Delta_{e,\min}$ and $f^*$ are redefined as:

$$\Delta_{e,\min} = \min_{A \in \Theta : e \in A, \Delta_A > 0} f(A, 1 - \bar{w}) - f(A^*, 1 - \bar{w}), \quad f^* = f(A^*, 1 - \bar{w}).$$

## 3 Analysis

We prove gap-dependent and gap-free upper bounds on the regret of `CombCascade` in Section 3.1. We discuss these bounds in Section 3.2.

### 3.1 Upper Bounds

We define the *suboptimality gap* of solution $A = (a_1, \ldots, a_{|A|})$ as $\Delta_A = f(A^*, \bar{w}) - f(A, \bar{w})$ and the probability that all items in $A$ are observed as $p_A = \prod_{k=1}^{|A|-1} \bar{w}(a_k)$. For convenience, we define

shorthands $f^* = f(A^*, \bar{w})$ and $p^* = p_{A^*}$. Let $\tilde{E} = E \setminus A^*$ be the set of *suboptimal items*, the items that are not in $A^*$. Then the *minimum gap* associated with suboptimal item $e \in \tilde{E}$ is:

$$\Delta_{e,\min} = f(A^*, \bar{w}) - \max_{A \in \Theta: e \in A, \Delta_A > 0} f(A, \bar{w}) \,.$$

Let $K = \max\{|A| : A \in \Theta\}$ be the maximum number of items in any solution and $f^* > 0$. Then the regret of `CombCascade` is bounded as follows.

**Theorem 1.** *The regret of* `CombCascade` *is bounded as* $R(n) \leq \dfrac{K}{f^*} \sum_{e \in \tilde{E}} \dfrac{4272}{\Delta_{e,\min}} \log n + \dfrac{\pi^2}{3} L.$

*Proof.* The proof is in Appendix A. The main idea is to reduce our analysis to that of `CombUCB1` in stochastic combinatorial semi-bandits [12]. This reduction is challenging for two reasons. First, our reward function is non-linear in the weights of chosen items. Second, we only observe some of the chosen items.

Our analysis can be trivially reduced to semi-bandits by conditioning on the event of observing all items. In particular, let $\mathcal{H}_t = (\mathbf{A}_1, \mathbf{O}_1, \ldots, \mathbf{A}_{t-1}, \mathbf{O}_{t-1}, \mathbf{A}_t)$ be the *history* of `CombCascade` up to choosing solution $\mathbf{A}_t$, the first $t-1$ observations and $t$ actions. Then we can express the expected regret at time $t$ conditioned on $\mathcal{H}_t$ as:

$$\mathbb{E}\left[R(\mathbf{A}_t, \mathbf{w}_t) \mid \mathcal{H}_t\right] = \mathbb{E}\left[\Delta_{\mathbf{A}_t}(1/p_{\mathbf{A}_t})\mathbb{1}\{\Delta_{\mathbf{A}_t} > 0, \mathbf{O}_t \geq |\mathbf{A}_t|\} \mid \mathcal{H}_t\right]$$

and analyze our problem under the assumption that all items in $\mathbf{A}_t$ are observed. This reduction is problematic because the probability $p_{\mathbf{A}_t}$ can be low, and as a result we get a loose regret bound.

We address this issue by formalizing the following insight into our problem. When $f(A, \bar{w}) \ll f^*$, `CombCascade` can distinguish $A$ from $A^*$ without learning the expected weights of all items in $A$. In particular, `CombCascade` acts implicitly on the prefixes of suboptimal solutions, and we choose them in our analysis such that the probability of observing all items in the prefixes is "close" to $f^*$, and the gaps are "close" to those of the original solutions.

**Lemma 1.** *Let* $A = (a_1, \ldots, a_{|A|}) \in \Theta$ *be a feasible solution and* $B_k = (a_1, \ldots, a_k)$ *be a* prefix *of* $k \leq |A|$ *items of* $A$. *Then* $k$ *can be set such that* $\Delta_{B_k} \geq \frac{1}{2}\Delta_A$ *and* $p_{B_k} \geq \frac{1}{2}f^*$.

Then we count the number of times that the prefixes can be chosen instead of $A^*$ when all items in the prefixes are observed. The last remaining issue is that $f(A, \mathbf{U}_t)$ is non-linear in the confidence radii of the items in $A$. Therefore, we bound it from above based on the following lemma.

**Lemma 2.** *Let* $0 \leq p_1, \ldots, p_K \leq 1$ *and* $u_1, \ldots, u_K \geq 0$. *Then:*

$$\prod_{k=1}^{K} \min\{p_k + u_k, 1\} \leq \prod_{k=1}^{K} p_k + \sum_{k=1}^{K} u_k \,.$$

*This bound is tight when* $p_1, \ldots, p_K = 1$ *and* $u_1, \ldots, u_K = 0$.

The rest of our analysis is along the lines of Theorem 5 in Kveton *et al.* [12]. We can achieve linear dependency on $K$, in exchange for a multiplicative factor of $534$ in our upper bound. ■

We also prove the following gap-free bound.

**Theorem 2.** *The regret of* `CombCascade` *is bounded as* $R(n) \leq 131\sqrt{\dfrac{KLn \log n}{f^*}} + \dfrac{\pi^2}{3} L.$

*Proof.* The proof is in Appendix B. The key idea is to decompose the regret of `CombCascade` into two parts, where the gaps $\Delta_{\mathbf{A}_t}$ are at most $\epsilon$ and larger than $\epsilon$. We analyze each part separately and then set $\epsilon$ to get the desired result. ■

### 3.2  Discussion

In Section 3.1, we prove two upper bounds on the $n$-step regret of `CombCascade`:

$$\text{Theorem 1: } O(KL(1/f^*)(1/\Delta) \log n)\,, \quad \text{Theorem 2: } O(\sqrt{KL(1/f^*)n \log n})\,,$$

where $\Delta = \min_{e \in \tilde{E}} \Delta_{e,\min}$. These bounds do not depend on the total number of feasible solutions $|\Theta|$ and are polynomial in any other quantity of interest. The bounds match, up to $O(1/f^*)$ factors,

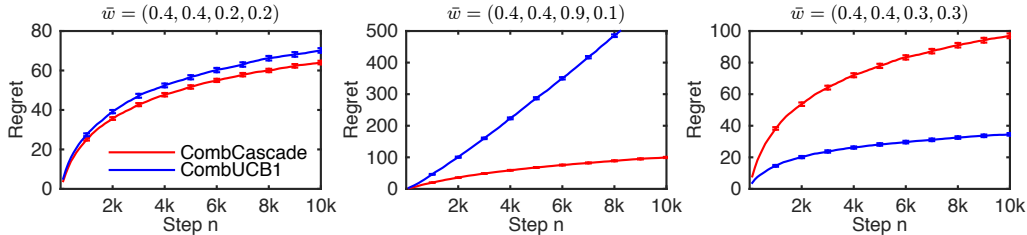

Figure 1: The regret of `CombCascade` and `CombUCB1` in the synthetic experiment (Section 4.1). The results are averaged over 100 runs.

the upper bounds of `CombUCB1` in stochastic combinatorial semi-bandits [12]. Since `CombCascade` receives less feedback than `CombUCB1`, this is rather surprising and unexpected. The upper bounds of Kveton *et al.* [12] are known to be tight up to polylogarithmic factors. We believe that our upper bounds are also tight in the setting similar to Kveton *et al.* [12], where the expected weight of each item is close to 1 and likely to be observed.

The assumption that $f^*$ is large is often reasonable. In network routing, the optimal routing path is likely to be reliable. In recommender systems, the optimal recommended list often does not satisfy a reasonably large fraction of users.

## 4 Experiments

We evaluate `CombCascade` in three experiments. In Section 4.1, we compare it to `CombUCB1` [12], a state-of-the-art algorithm for stochastic combinatorial semi-bandits with a linear reward function. This experiment shows that `CombUCB1` cannot solve all instances of our problem, which highlights the need for a new learning algorithm. It also shows the limitations of `CombCascade`. We evaluate `CombCascade` on two real-world problems in Sections 4.2 and 4.3.

### 4.1 Synthetic

In the first experiment, we compare `CombCascade` to `CombUCB1` [12] on a synthetic problem. This problem is a combinatorial cascading bandit with $L = 4$ items and $\Theta = \{(1, 2), (3, 4)\}$. `CombUCB1` is a popular algorithm for stochastic combinatorial semi-bandits with a linear reward function. We approximate $\max_{A \in \Theta} f(A, w)$ by $\min_{A \in \Theta} \sum_{e \in A}(1 - w(e))$. This approximation is motivated by the fact that $f(A, w) = \prod_{e \in A} w(e) \approx 1 - \sum_{e \in A}(1 - w(e))$ as $\min_{e \in E} w(e) \to 1$. We update the estimates of $\bar{w}$ in `CombUCB1` as in `CombCascade`, based on the weights of the observed items in (1).

We experiment with three different settings of $\bar{w}$ and report our results in Figure 1. The settings of $\bar{w}$ are reported in our plots. We assume that $\mathbf{w}_t(e)$ are distributed independently, except for the last plot where $\mathbf{w}_t(3) = \mathbf{w}_t(4)$. Our plots represent three common scenarios that we encountered in our experiments. In the first plot, $\arg\max_{A \in \Theta} f(A, \bar{w}) = \arg\min_{A \in \Theta} \sum_{e \in A}(1 - \bar{w}(e))$. In this case, both `CombCascade` and `CombUCB1` can learn $A^*$. The regret of `CombCascade` is slightly lower than that of `CombUCB1`. In the second plot, $\arg\max_{A \in \Theta} f(A, \bar{w}) \neq \arg\min_{A \in \Theta} \sum_{e \in A}(1 - \bar{w}(e))$. In this case, `CombUCB1` cannot learn $A^*$ and therefore suffers linear regret. In the third plot, we violate our modeling assumptions. Perhaps surprisingly, `CombCascade` can still learn the optimal solution $A^*$, although it suffers higher regret than `CombUCB1`.

### 4.2 Network Routing

In the second experiment, we evaluate `CombCascade` on a problem of network routing. We experiment with six networks from the *RocketFuel* dataset [17], which are described in Figure 2a.

Our learning problem is formulated as follows. The ground set $E$ are the links in the network. The feasible set $\Theta$ are all paths in the network. At time $t$, we generate a random pair of *starting* and *end nodes*, and the learning agent chooses a routing path between these nodes. The goal of the agent is to maximizes the probability that all links in the path are *up*. The feedback is the index of the first link in the path which is *down*. The weight of link $e$ at time $t$, $\mathbf{w}_t(e)$, is an indicator of link $e$ being

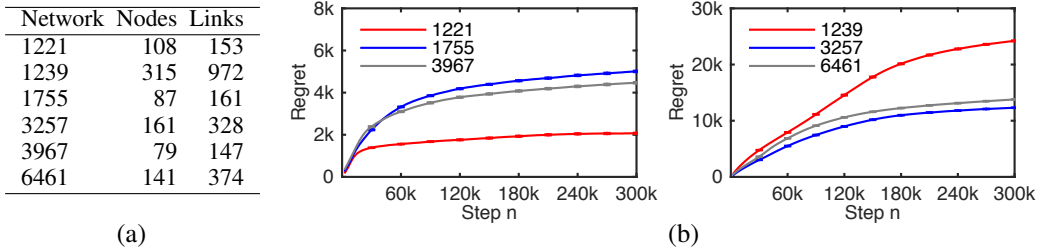

Figure 2: **a**. The description of six networks from our network routing experiment (Section 4.2). **b**. The $n$-step regret of `CombCascade` in these networks. The results are averaged over 50 runs.

*up* at time $t$. We model $\mathbf{w}_t(e)$ as an independent Bernoulli random variable $\mathbf{w}_t(e) \sim \mathrm{B}(\bar{\mathbf{w}}(e))$ with mean $\bar{\mathbf{w}}(e) = 0.7 + 0.2\,\mathrm{local}(e)$, where $\mathrm{local}(e)$ is an indicator of link $e$ being local. We say that the link is *local* when its expected latency is at most 1 millisecond. About a half of the links in our networks are local. To summarize, the local links are up with probability 0.9; and are more reliable than the global links, which are up only with probability 0.7.

Our results are reported in Figure 2b. We observe that the $n$-step regret of `CombCascade` flattens as time $n$ increases. This means that `CombCascade` learns near-optimal policies in all networks.

### 4.3 Diverse Recommendations

In our last experiment, we evaluate `CombCascade` on a problem of diverse recommendations. This problem is motivated by on-demand media streaming services like Netflix, which often recommend groups of movies, such as "Popular on Netflix" and "Dramas". We experiment with the *MovieLens* dataset [13] from March 2015. The dataset contains 138k people who assigned 20M ratings to 27k movies between January 1995 and March 2015.

Our learning problem is formulated as follows. The ground set $E$ are 200 movies from our dataset: 25 most rated animated movies, 75 random animated movies, 25 most rated non-animated movies, and 75 random non-animated movies. The feasible set $\Theta$ are all $K$-permutations of $E$ where $K/2$ movies are animated. The weight of item $e$ at time $t$, $\mathbf{w}_t(e)$, indicates that item $e$ attracts the user at time $t$. We assume that $\mathbf{w}_t(e) = 1$ if and only if the user rated item $e$ in our dataset. This indicates that the user watched movie $e$ at some point in time, perhaps because the movie was attractive. The user at time $t$ is drawn randomly from our pool of users. The goal of the learning agent is to learn a list of items $A^* = \arg\max_{A \in \Theta} \mathbb{E}\left[f_\vee(A, \mathbf{w})\right]$ that maximizes the probability that at least one item is attractive. The feedback is the index of the first attractive item in the list (Section 2.3). We would like to point out that our modeling assumptions are violated in this experiment. In particular, $\mathbf{w}_t(e)$ are correlated across items $e$ because the users do not rate movies independently. The result is that $A^* \neq \arg\max_{A \in \Theta} f_\vee(A, \bar{w})$. It is NP-hard to compute $A^*$. However, $\mathbb{E}\left[f_\vee(A, \mathbf{w})\right]$ is submodular and monotone in $A$, and therefore a $(1 - 1/e)$ approximation to $A^*$ can be computed greedily. We denote this approximation by $A^*$ and show it for $K = 8$ in Figure 3a.

Our results are reported in Figure 3b. Similarly to Figure 2b, the $n$-step regret of `CombCascade` is a concave function of time $n$ for all studied $K$. This indicates that `CombCascade` solutions improve over time. We note that the regret does not flatten as in Figure 2b. The reason is that `CombCascade` does not learn $A^*$. Nevertheless, it performs well and we expect comparably good performance in other domains where our modeling assumptions are not satisfied. Our current theory cannot explain this behavior and we leave it for future work.

## 5 Related Work

Our work generalizes cascading bandits of Kveton *et al.* [10] to arbitrary combinatorial constraints. The feasible set in cascading bandits is a uniform matroid, any list of $K$ items out of $L$ is feasible. Our generalization significantly expands the applicability of the original model and we demonstrate this on two novel real-world problems (Section 4). Our work also extends stochastic combinatorial semi-bandits with a linear reward function [8, 11, 12] to the cascade model of feedback. A similar model to cascading bandits was recently studied by Combes *et al.* [7].

| Movie title | Animation |
|---|---|
| Pulp Fiction | No |
| Forrest Gump | No |
| Independence Day | No |
| Shawshank Redemption | No |
| Toy Story | Yes |
| Shrek | Yes |
| Who Framed Roger Rabbit? | Yes |
| Aladdin | Yes |

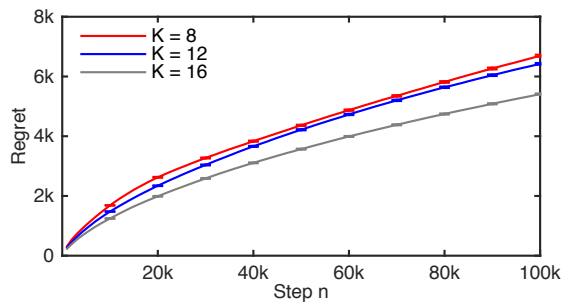

(a)                                                           (b)

Figure 3: **a**. The optimal list of $8$ movies in the diverse recommendations experiment (Section 4.3). **b**. The $n$-step regret of `CombCascade` in this experiment. The results are averaged over 50 runs.

Our generalization is significant for two reasons. First, `CombCascade` is a novel learning algorithm. `CombUCB1` [12] chooses solutions with the largest *sum* of the UCBs. `CascadeUCB1` [10] chooses $K$ items out of $L$ with the largest UCBs. `CombCascade` chooses solutions with the largest *product* of the UCBs. All three algorithms can find the optimal solution in cascading bandits. However, when the feasible set is not a matroid, it is critical to maximize the product of the UCBs. `CombUCB1` may learn a suboptimal solution in this setting and we illustrate this in Section 4.1.

Second, our analysis is novel. The proof of Theorem 1 is different from those of Theorems 2 and 3 in Kveton *et al.* [10]. These proofs are based on counting the number of times that each suboptimal item is chosen instead of any optimal item. They can be only applied to special feasible sets, such a matroid, because they require that the items in the feasible solutions are exchangeable. We build on the recent work of Kveton *et al.* [12] to achieve linear dependency on $K$ in Theorem 1. The rest of our analysis is novel.

Our problem is a partial monitoring problem where some of the chosen items may be unobserved. Agrawal *et al.* [1] and Bartok *et al.* [4] studied partial monitoring problems and proposed learning algorithms for solving them. These algorithms are impractical in our setting. As an example, if we formulate our problem as in Bartok *et al.* [4], we get $|\Theta|$ actions and $2^L$ unobserved outcomes; and the learning algorithm reasons over $|\Theta|^2$ pairs of actions and requires $O(2^L)$ space. Lin *et al.* [15] also studied combinatorial partial monitoring. Their feedback is a linear function of the weights of chosen items. Our feedback is a non-linear function of the weights.

Our reward function is non-linear in unknown parameters. Chen *et al.* [5] studied stochastic combinatorial semi-bandits with a non-linear reward function, which is a known monotone function of an unknown linear function. The feedback in Chen *et al.* [5] is semi-bandit, which is more informative than in our work. Le *et al.* [14] studied a network optimization problem where the reward function is a non-linear function of observations.

## 6   Conclusions

We propose combinatorial cascading bandits, a class of stochastic partial monitoring problems that can model many practical problems, such as learning of a routing path in an unreliable communication network that maximizes the probability of packet delivery, and learning to recommend a list of attractive items. We propose a practical UCB-like algorithm for our problems, `CombCascade`, and prove upper bounds on its regret. We evaluate `CombCascade` on two real-world problems and show that it performs well even when our modeling assumptions are violated.

Our results and analysis apply to any combinatorial action set, and therefore are quite general. The strongest assumption in our work is that the weights of items are distributed independently of each other. This assumption is critical and hard to eliminate (Section 2.1). Nevertheless, it can be easily relaxed to conditional independence given the features of items, along the lines of Wen *et al.* [19]. We leave this for future work. From the theoretical point of view, we want to derive a lower bound on the $n$-step regret in combinatorial cascading bandits, and show that the factor of $f^*$ in Theorems 1 and 2 is intrinsic.

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
