[Supplementary Material]

# A   Proof of Theorem 1

Our proof has four main parts. In Appendix A.1, we bound the regret associated with the event that our high-probability confidence intervals do not hold. In Appendix A.2, we change counted events, from partially-observed suboptimal solutions to their fully-observed prefixes. In Appendix A.3, we bound the number of times that any suboptimal prefix can be chosen instead of the optimal solution $A^*$. In Appendix A.4, we apply the counting argument of Kveton *et al.* [12] and finish our proof.

Let $\mathbf{R}_t = R(\mathbf{A}_t, \mathbf{w}_t)$ be the stochastic regret of CombCascade at time $t$, where $\mathbf{A}_t$ and $\mathbf{w}_t$ are the solution and the weights of the items at time $t$, respectively. Let:

$$\mathcal{E}_t = \left\{ \exists e \in E \text{ s.t. } \left| \bar{w}(e) - \hat{\mathbf{w}}_{\mathbf{T}_{t-1}(e)}(e) \right| \geq c_{t-1, \mathbf{T}_{t-1}(e)} \right\}$$

be the event that $\bar{w}(e)$ is outside of the high-probability confidence interval around $\hat{\mathbf{w}}_{\mathbf{T}_{t-1}(e)}(e)$ for at least one item $e \in E$ at time $t$; and let $\overline{\mathcal{E}}_t$ be the complement of event $\mathcal{E}_t$, the event that $\bar{w}(e)$ is in the high-probability confidence interval around $\hat{\mathbf{w}}_{\mathbf{T}_{t-1}(e)}(e)$ for all items $e \in E$ at time $t$. Then we can decompose the expected regret of CombCascade as:

$$R(n) = \mathbb{E}\left[ \sum_{t=1}^{n} \mathbb{1}\{\mathcal{E}_t\} \mathbf{R}_t \right] + \mathbb{E}\left[ \sum_{t=1}^{n} \mathbb{1}\{\overline{\mathcal{E}}_t\} \mathbf{R}_t \right]. \tag{4}$$

## A.1   Confidence Intervals Fail

The first term in (4) is easy to bound because $\mathbf{R}_t$ is bounded and our confidence intervals hold with high probability. In particular, Hoeffding's inequality yields that for any $e$, $s$, and $t$:

$$P(|\bar{w}(e) - \hat{\mathbf{w}}_s(e)| \geq c_{t,s}) \leq 2\exp[-3\log t],$$

and therefore:

$$\mathbb{E}\left[ \sum_{t=1}^{n} \mathbb{1}\{\mathcal{E}_t\} \right] \leq \sum_{e \in E} \sum_{t=1}^{n} \sum_{s=1}^{t} P(|\bar{w}(e) - \hat{\mathbf{w}}_s(e)| \geq c_{t,s})$$

$$\leq 2 \sum_{e \in E} \sum_{t=1}^{n} \sum_{s=1}^{t} \exp[-3\log t] \leq 2 \sum_{e \in E} \sum_{t=1}^{n} t^{-2} \leq \frac{\pi^2}{3} L.$$

Since $\mathbf{R}_t \leq 1$, $\mathbb{E}\left[ \sum_{t=1}^{n} \mathbb{1}\{\mathcal{E}_t\} \mathbf{R}_t \right] \leq \frac{\pi^2}{3} L$.

## A.2   From Partially-Observed Solutions to Fully-Observed Prefixes

Let $\mathcal{H}_t = (\mathbf{A}_1, \mathbf{O}_1, \ldots, \mathbf{A}_{t-1}, \mathbf{O}_{t-1}, \mathbf{A}_t)$ be the *history* of CombCascade up to choosing solution $\mathbf{A}_t$, the first $t-1$ observations and $t$ actions. Let $\mathbb{E}[\cdot \mid \mathcal{H}_t]$ be the conditional expectation given this history. Then we can rewrite the expected regret at time $t$ conditioned on $\mathcal{H}_t$ as:

$$\mathbb{E}[\mathbf{R}_t \mid \mathcal{H}_t] = \mathbb{E}[\Delta_{\mathbf{A}_t} \mathbb{1}\{\Delta_{\mathbf{A}_t} > 0\} \mid \mathcal{H}_t] = \mathbb{E}\left[ \frac{\Delta_{\mathbf{A}_t}}{p_{\mathbf{A}_t}} \mathbb{1}\{\Delta_{\mathbf{A}_t} > 0, \ \mathbf{O}_t \geq |\mathbf{A}_t|\} \ \middle| \ \mathcal{H}_t \right]$$

and analyze our problem under the assumption that all items in $\mathbf{A}_t$ are observed. This reduction is problematic because the probability $p_{\mathbf{A}_t}$ can be low, and as a result we get a loose regret bound. To address this problem, we introduce the notion of prefixes.

Let $A = (a_1, \ldots, a_{|A|})$. Then $B = (a_1, \ldots, a_k)$ is a *prefix* of $A$ for any $k \leq |A|$. In the rest of our analysis, we treat prefixes as feasible solutions to our original problem. Let $\mathbf{B}_t$ be a prefix of $\mathbf{A}_t$ as defined in Lemma 1. Then $\Delta_{\mathbf{B}_t} \geq \frac{1}{2} \Delta_{\mathbf{A}_t}$ and $p_{\mathbf{B}_t} \geq \frac{1}{2} f^*$, and we can bound the expected regret at time $t$ conditioned on $\mathcal{H}_t$ as:

$$\mathbb{E}[\mathbf{R}_t \mid \mathcal{H}_t] = \mathbb{E}\left[ \frac{\Delta_{\mathbf{A}_t}}{p_{\mathbf{B}_t}} \mathbb{1}\{\Delta_{\mathbf{A}_t} > 0, \ \mathbf{O}_t \geq |\mathbf{B}_t|\} \ \middle| \ \mathcal{H}_t \right]$$

$$\leq \frac{4}{f^*} \mathbb{E}[\Delta_{\mathbf{B}_t} \mathbb{1}\{\Delta_{\mathbf{B}_t} > 0, \ \mathbf{O}_t \geq |\mathbf{B}_t|\} \mid \mathcal{H}_t]. \tag{5}$$

Now we bound the second term in (4):

$$\mathbb{E}\left[\sum_{t=1}^{n} \mathbb{1}\{\overline{\mathcal{E}}_t\} \mathbf{R}_t\right] \overset{(a)}{=} \sum_{t=1}^{n} \mathbb{E}\left[\mathbb{1}\{\overline{\mathcal{E}}_t\} \mathbb{E}\left[\mathbf{R}_t \mid \mathcal{H}_t\right]\right]$$

$$\overset{(b)}{\leq} \frac{4}{f^*} \mathbb{E}\left[\sum_{t=1}^{n} \Delta_{\mathbf{B}_t} \mathbb{1}\{\overline{\mathcal{E}}_t,\ \Delta_{\mathbf{B}_t} > 0,\ \mathbf{O}_t \geq |\mathbf{B}_t|\}\right]. \tag{6}$$

Equality (a) is due to the tower rule and that $\mathbb{1}\{\overline{\mathcal{E}}_t\}$ is only a function of $\mathcal{H}_t$. Inequality (b) follows from the upper bound in (5).

## A.3 Counting Suboptimal Prefixes

Let:

$$\mathcal{F}_t = \left\{2 \sum_{e \in \tilde{\mathbf{B}}_t} c_{n, \mathbf{T}_{t-1}(e)} \geq \Delta_{\mathbf{B}_t},\ \Delta_{\mathbf{B}_t} > 0,\ \mathbf{O}_t \geq |\mathbf{B}_t|\right\} \tag{7}$$

be the event that suboptimal prefix $\mathbf{B}_t$ is "hard to distinguish" from $A^*$, where $\tilde{\mathbf{B}}_t = \mathbf{B}_t \setminus A^*$ is the set of suboptimal items in $\mathbf{B}_t$. The goal of this section is to bound (6) by a function of $\mathcal{F}_t$.

We bound $\Delta_{\mathbf{B}_t} \mathbb{1}\{\overline{\mathcal{E}}_t,\ \Delta_{\mathbf{B}_t} > 0,\ \mathbf{O}_t \geq |\mathbf{B}_t|\}$ from above for any suboptimal prefix $\mathbf{B}_t$. Our bound is proved based on several facts. First, $\mathbf{B}_t$ is a prefix of $\mathbf{A}_t$, and hence $f(\mathbf{B}_t, \mathbf{U}_t) \geq f(\mathbf{A}_t, \mathbf{U}_t)$ for any $\mathbf{U}_t$. Second, when CombCascade chooses $\mathbf{A}_t$, $f(\mathbf{A}_t, \mathbf{U}_t) \geq f(A^*, \mathbf{U}_t)$. It follows that:

$$\prod_{e \in \mathbf{B}_t} \mathbf{U}_t(e) = f(\mathbf{B}_t, \mathbf{U}_t) \geq f(\mathbf{A}_t, \mathbf{U}_t) \geq f(A^*, \mathbf{U}_t) = \prod_{e \in A^*} \mathbf{U}_t(e).$$

Now we divide both sides by $\prod_{e \in A^* \cap \mathbf{B}_t} \mathbf{U}_t(e)$:

$$\prod_{e \in \tilde{\mathbf{B}}_t} \mathbf{U}_t(e) \geq \prod_{e \in A^* \setminus \mathbf{B}_t} \mathbf{U}_t(e)$$

and substitute the definitions of the UCBs from (3):

$$\prod_{e \in \tilde{\mathbf{B}}_t} \min\left\{\hat{\mathbf{w}}_{\mathbf{T}_{t-1}(e)}(e) + c_{t-1, \mathbf{T}_{t-1}(e)}, 1\right\} \geq \prod_{e \in A^* \setminus \mathbf{B}_t} \min\left\{\hat{\mathbf{w}}_{\mathbf{T}_{t-1}(e)}(e) + c_{t-1, \mathbf{T}_{t-1}(e)}, 1\right\}.$$

Since $\overline{\mathcal{E}}_t$ happens, $\left|\bar{w}(e) - \hat{\mathbf{w}}_{\mathbf{T}_{t-1}(e)}(e)\right| < c_{t-1, \mathbf{T}_{t-1}(e)}$ for all $e \in E$ and therefore:

$$\prod_{e \in A^* \setminus \mathbf{B}_t} \min\left\{\hat{\mathbf{w}}_{\mathbf{T}_{t-1}(e)}(e) + c_{t-1, \mathbf{T}_{t-1}(e)}, 1\right\} \geq \prod_{e \in A^* \setminus \mathbf{B}_t} \bar{w}(e)$$

$$\prod_{e \in \tilde{\mathbf{B}}_t} \min\left\{\hat{\mathbf{w}}_{\mathbf{T}_{t-1}(e)}(e) + c_{t-1, \mathbf{T}_{t-1}(e)}, 1\right\} \leq \prod_{e \in \tilde{\mathbf{B}}_t} \min\left\{\bar{w}(e) + 2c_{t-1, \mathbf{T}_{t-1}(e)}, 1\right\}.$$

By Lemma 2:

$$\prod_{e \in \tilde{\mathbf{B}}_t} \min\left\{\bar{w}(e) + 2c_{t-1, \mathbf{T}_{t-1}(e)}, 1\right\} \leq \prod_{e \in \tilde{\mathbf{B}}_t} \bar{w}(e) + 2 \sum_{e \in \tilde{\mathbf{B}}_t} c_{t-1, \mathbf{T}_{t-1}(e)}.$$

Finally, we chain the last four inequalities and get:

$$\prod_{e \in \tilde{\mathbf{B}}_t} \bar{w}(e) + 2 \sum_{e \in \tilde{\mathbf{B}}_t} c_{t-1, \mathbf{T}_{t-1}(e)} \geq \prod_{e \in A^* \setminus \mathbf{B}_t} \bar{w}(e),$$

which further implies that:

$$2 \sum_{e \in \tilde{\mathbf{B}}_t} c_{t-1, \mathbf{T}_{t-1}(e)} \geq \prod_{e \in A^* \setminus \mathbf{B}_t} \bar{w}(e) - \prod_{e \in \tilde{\mathbf{B}}_t} \bar{w}(e)$$

$$\geq \underbrace{\prod_{e \in A^* \cap \mathbf{B}_t} \bar{w}(e)}_{\leq 1} \left[\prod_{e \in A^* \setminus \mathbf{B}_t} \bar{w}(e) - \prod_{e \in \tilde{\mathbf{B}}_t} \bar{w}(e)\right]$$

$$= \Delta_{\mathbf{B}_t}.$$

Since $c_{n, \mathbf{T}_{t-1}(e)} \geq c_{t-1, \mathbf{T}_{t-1}(e)}$ for any time $t \leq n$, the event $\mathcal{F}_t$ in (7) happens. Therefore, we can bound the right-hand side in (6) as:

$$\mathbb{E}\left[\sum_{t=1}^{n} \Delta_{\mathbf{B}_t} \mathbb{1}\{\overline{\mathcal{E}}_t, \, \Delta_{\mathbf{B}_t} > 0, \, \mathbf{O}_t \geq |\mathbf{B}_t|\}\right] \leq \mathbb{E}\left[\hat{\mathbf{R}}(n)\right],$$

where:

$$\hat{\mathbf{R}}(n) = \sum_{t=1}^{n} \Delta_{\mathbf{B}_t} \mathbb{1}\{\mathcal{F}_t\}. \tag{8}$$

### A.4 CombUCB1 **Analysis of Kveton** *et al.* **[12]**

It remains to bound $\hat{\mathbf{R}}(n)$ in (8). Note that the event $\mathcal{F}_t$ can happen only if the weights of all items in $\mathbf{B}_t$ are observed. As a result, $\hat{\mathbf{R}}(n)$ can be bounded as in stochastic combinatorial semi-bandits. The key idea of our proof is to introduce infinitely-many mutually-exclusive events and then bound the number of times that these events happen when a suboptimal prefix is chosen [12]. The event $i$ at time $t$ is:

$$G_{i,t} = \{\text{less than } \beta_1 K \text{ items in } \tilde{\mathbf{B}}_t \text{ were observed at most } \alpha_1 \frac{K^2}{\Delta_{\mathbf{B}_t}^2} \log n \text{ times,}$$

$$\ldots,$$

$$\text{less than } \beta_{i-1} K \text{ items in } \tilde{\mathbf{B}}_t \text{ were observed at most } \alpha_{i-1} \frac{K^2}{\Delta_{\mathbf{B}_t}^2} \log n \text{ times,}$$

$$\text{at least } \beta_i K \text{ items in } \tilde{\mathbf{B}}_t \text{ were observed at most } \alpha_i \frac{K^2}{\Delta_{\mathbf{B}_t}^2} \log n \text{ times,}$$

$$\mathbf{O}_t \geq |\mathbf{B}_t|\},$$

where we assume that $\Delta_{\mathbf{B}_t} > 0$; and the constants $(\alpha_i)$ and $(\beta_i)$ are defined as:

$$1 = \beta_0 > \beta_1 > \beta_2 > \ldots > \beta_k > \ldots$$
$$\alpha_1 > \alpha_2 > \ldots > \alpha_k > \ldots,$$

and satisfy $\lim_{i \to \infty} \alpha_i = \lim_{i \to \infty} \beta_i = 0$. By Lemma 3 of Kveton *et al.* [12], $G_{i,t}$ are exhaustive at any time $t$ when $(\alpha_i)$ and $(\beta_i)$ satisfy:

$$\sqrt{6} \sum_{i=1}^{\infty} \frac{\beta_{i-1} - \beta_i}{\sqrt{\alpha_i}} \leq 1.$$

In this case:

$$\hat{\mathbf{R}}(n) = \sum_{t=1}^{n} \Delta_{\mathbf{B}_t} \mathbb{1}\{\mathcal{F}_t\} = \sum_{i=1}^{\infty} \sum_{t=1}^{n} \Delta_{\mathbf{B}_t} \mathbb{1}\{G_{i,t}, \, \Delta_{\mathbf{B}_t} > 0\}.$$

Now we introduce item-specific variants of events $G_{i,t}$ and associate the regret at time $t$ with these events. In particular, let:

$$G_{e,i,t} = G_{i,t} \cap \left\{e \in \tilde{\mathbf{B}}_t, \, \mathbf{T}_{t-1}(e) \leq \alpha_i \frac{K^2}{\Delta_{\mathbf{B}_t}^2} \log n\right\}$$

be the event that item $e$ is not observed "sufficiently often" under event $G_{i,t}$. Then it follows that:

$$\mathbb{1}\{G_{i,t}, \, \Delta_{\mathbf{B}_t} > 0\} \leq \frac{1}{\beta_i K} \sum_{e \in \tilde{E}} \mathbb{1}\{G_{e,i,t}, \, \Delta_{\mathbf{B}_t} > 0\}$$

because at least $\beta_i K$ items are not observed "sufficiently often" under event $G_{i,t}$. Therefore, we can bound $\hat{\mathbf{R}}(n)$ as:

$$\hat{\mathbf{R}}(n) \leq \sum_{e \in \tilde{E}} \sum_{i=1}^{\infty} \sum_{t=1}^{n} \mathbb{1}\{G_{e,i,t}, \, \Delta_{\mathbf{B}_t} > 0\} \frac{\Delta_{\mathbf{B}_t}}{\beta_i K}.$$

Let each item $e$ be in $N_e$ suboptimal prefixes and $\Delta_{e,1} \geq \ldots \geq \Delta_{e,N_e}$ be the gaps of these prefixes, ordered from the largest gap to the smallest. Then $\hat{\mathbf{R}}(n)$ can be further bounded as:

$$\hat{\mathbf{R}}(n) \leq \sum_{e \in \tilde{E}} \sum_{i=1}^{\infty} \sum_{t=1}^{n} \sum_{k=1}^{N_e} \mathbb{1}\{G_{e,i,t}, \ \Delta_{\mathbf{B}_t} = \Delta_{e,k}\} \frac{\Delta_{e,k}}{\beta_i K}$$

$$\overset{(a)}{\leq} \sum_{e \in \tilde{E}} \sum_{i=1}^{\infty} \sum_{t=1}^{n} \sum_{k=1}^{N_e} \mathbb{1}\left\{e \in \tilde{\mathbf{B}}_t, \ \mathbf{T}_{t-1}(e) \leq \alpha_i \frac{K^2}{\Delta_{e,k}^2} \log n, \ \Delta_{\mathbf{B}_t} = \Delta_{e,k}, \ \mathbf{O}_t \geq |\mathbf{B}_t|\right\} \frac{\Delta_{e,k}}{\beta_i K}$$

$$\overset{(b)}{\leq} \sum_{e \in \tilde{E}} \sum_{i=1}^{\infty} \frac{\alpha_i K \log n}{\beta_i} \left[\Delta_{e,1} \frac{1}{\Delta_{e,1}^2} + \sum_{k=2}^{N_e} \Delta_{e,k}\left(\frac{1}{\Delta_{e,k}^2} - \frac{1}{\Delta_{e,k-1}^2}\right)\right]$$

$$\overset{(c)}{<} \sum_{e \in \tilde{E}} \sum_{i=1}^{\infty} \frac{\alpha_i K \log n}{\beta_i} \frac{2}{\Delta_{e,N_e}}$$

$$= \sum_{e \in \tilde{E}} K \frac{2}{\Delta_{e,N_e}} \left[\sum_{i=1}^{\infty} \frac{\alpha_i}{\beta_i}\right] \log n \,,$$

where inequality (a) follows from the definition of $G_{e,i,t}$ and inequality (b) is from solving:

$$\max_{A_{1:n}, O_{1:n}} \sum_{t=1}^{n} \sum_{k=1}^{N_e} \mathbb{1}\left\{e \in \tilde{B}_t, \ T_{t-1}^{A_{1:n}, O_{1:n}}(e) \leq \alpha_i \frac{K^2}{\Delta_{e,k}^2} \log n, \ \Delta_{B_t} = \Delta_{e,k}, \ O_t \geq |B_t|\right\} \frac{\Delta_{e,k}}{\beta_i K} \,,$$

where $A_{1:n} = (A_1, \ldots, A_n)$ is a sequence of $n$ solutions, $O_{1:n} = (O_1, \ldots, O_n)$ is a sequence of $n$ observations, $T_t^{A_{1:n}, O_{1:n}}(e)$ is the number of times that item $e$ is observed in $t$ steps under $A_{1:n}$ and $O_{1:n}$, $B_t$ is the prefix of $A_t$ as defined in Lemma 1, and $\tilde{B}_t = B_t \setminus A^*$. Inequality (c) is by Lemma 3 of Kveton *et al.* [11]:

$$\left[\Delta_{e,1} \frac{1}{\Delta_{e,1}^2} + \sum_{k=2}^{N_e} \Delta_{e,k}\left(\frac{1}{\Delta_{e,k}^2} - \frac{1}{\Delta_{e,k-1}^2}\right)\right] < \frac{2}{\Delta_{e,N_e}} \,.$$

For the same $(\alpha_i)$ and $(\beta_i)$ as in Theorem 4 of Kveton *et al.* [12], $\sum_{i=1}^{\infty} \frac{\alpha_i}{\beta_i} < 267$. Moreover, since $\Delta_{\mathbf{B}_t} \geq \frac{1}{2}\Delta_{\mathbf{A}_t}$ for any solution $\mathbf{A}_t$ and its prefix $\mathbf{B}_t$, we have $\Delta_{e,N_e} \geq \frac{1}{2}\Delta_{e,\min}$. Now we chain all inequalities and get:

$$R(n) \leq \frac{4}{f^*} \mathbb{E}\left[\hat{\mathbf{R}}(n)\right] + \frac{\pi^2}{3} L \leq \frac{K}{f^*} \sum_{e \in \tilde{E}} \frac{4272}{\Delta_{e,\min}} \log n + \frac{\pi^2}{3} L \,.$$

# B   Proof of Theorem 2

The key idea is to decompose the regret of CombCascade into two parts, where the gaps $\Delta_{\mathbf{A}_t}$ are at most $\epsilon$ and larger than $\epsilon$. In particular, note that for any $\epsilon > 0$:

$$R(n) = \mathbb{E}\left[\sum_{t=1}^{n} \mathbb{1}\{\Delta_{\mathbf{A}_t} \leq \varepsilon\} \mathbf{R}_t\right] + \mathbb{E}\left[\sum_{t=1}^{n} \mathbb{1}\{\Delta_{\mathbf{A}_t} > \varepsilon\} \mathbf{R}_t\right] \,. \tag{9}$$

The first term in (9) can be bounded trivially as:

$$\mathbb{E}\left[\sum_{t=1}^{n} \mathbb{1}\{\Delta_{\mathbf{A}_t} \leq \varepsilon\} \mathbf{R}_t\right] = \mathbb{E}\left[\sum_{t=1}^{n} \Delta_{\mathbf{A}_t} \mathbb{1}\{\Delta_{\mathbf{A}_t} \leq \varepsilon, \ \Delta_{\mathbf{A}_t} > 0\}\right] \leq \epsilon n$$

because $\Delta_{\mathbf{A}_t} \leq \varepsilon$. The second term in (9) can be bounded in the same way as $R(n)$ in Theorem 1. The only difference is that $\Delta_{e,\min} \geq \epsilon$ for all $e \in \tilde{E}$. Therefore:

$$\mathbb{E}\left[\sum_{t=1}^{n} \mathbb{1}\{\Delta_{\mathbf{A}_t} > \varepsilon\} \mathbf{R}_t\right] \leq \frac{K}{f^*} \sum_{e \in \tilde{E}} \frac{4272}{\Delta_{e,\min}} \log n + \frac{\pi^2}{3} L \leq \frac{4272KL}{f^*\epsilon} \log n + \frac{\pi^2}{3} L \,.$$

Now we chain all inequalities and get:

$$R(n) \leq \frac{4272KL}{f^*\epsilon} \log n + \epsilon n + \frac{\pi^2}{3}L \,.$$

Finally, we choose $\epsilon = \sqrt{\dfrac{4272KL \log n}{f^* n}}$ and get:

$$R(n) \leq 2\sqrt{4272}\sqrt{\frac{KLn \log n}{f^*}} + \frac{\pi^2}{3}L < 131\sqrt{\frac{KLn \log n}{f^*}} + \frac{\pi^2}{3}L \,,$$

which concludes our proof.

## C    Technical Lemmas

**Lemma 1.** *Let $A = (a_1, \ldots, a_{|A|}) \in \Theta$ be a feasible solution and $B_k = (a_1, \ldots, a_k)$ be a* prefix *of $k \leq |A|$ items of A. Then k can be set such that $\Delta_{B_k} \geq \frac{1}{2}\Delta_A$ and $p_{B_k} \geq \frac{1}{2}f^*$.*

*Proof.* We consider two cases. First, suppose that $f(A, \bar{w}) \geq \frac{1}{2}f^*$. Then our claims hold trivially for $k = |A|$. Now suppose that $f(A, \bar{w}) < \frac{1}{2}f^*$. Then we choose $k$ such that:

$$f(B_k, \bar{w}) \leq \frac{1}{2}f^* \leq p_{B_k} \,.$$

Such $k$ is guaranteed to exist because $\bigcup_{i=1}^{|A|}[f(B_i, \bar{w}), p_{B_i}] = [f(A, \bar{w}), 1]$, which follows from the facts that $f(B_i, \bar{w}) = p_{B_i}\bar{w}(a_i)$ for any $i \leq |A|$ and $p_{B_1} = 1$. We prove that $\Delta_{B_k} \geq \frac{1}{2}\Delta_A$ as:

$$\Delta_{B_k} = f^* - f(B_k, \bar{w}) \geq \frac{1}{2}f^* \geq \frac{1}{2}\Delta_A \,.$$

The first inequality is by our assumption and the second one holds for any solution $A$. ∎

**Lemma 2.** *Let $0 \leq p_1, \ldots, p_K \leq 1$ and $u_1, \ldots, u_K \geq 0$. Then:*

$$\prod_{k=1}^{K} \min\{p_k + u_k, 1\} \leq \prod_{k=1}^{K} p_k + \sum_{k=1}^{K} u_k \,.$$

*This bound is tight when $p_1, \ldots, p_K = 1$ and $u_1, \ldots, u_K = 0$.*

*Proof.* The proof is by induction on $K$. Our claim clearly holds when $K = 1$. Now choose $K > 1$ and suppose that our claim holds for any $0 \leq p_1, \ldots, p_{K-1} \leq 1$ and $u_1, \ldots, u_{K-1} \geq 0$. Then:

$$\prod_{k=1}^{K} \min\{p_k + u_k, 1\} = \min\{p_K + u_K, 1\} \prod_{k=1}^{K-1} \min\{p_k + u_k, 1\}$$

$$\leq \min\{p_K + u_K, 1\} \left( \prod_{k=1}^{K-1} p_k + \sum_{k=1}^{K-1} u_k \right)$$

$$\leq p_K \prod_{k=1}^{K-1} p_k + u_K \underbrace{\prod_{k=1}^{K-1} p_k}_{\leq 1} + \underbrace{\min\{p_K + u_K, 1\}}_{\leq 1} \sum_{k=1}^{K-1} u_k$$

$$\leq \prod_{k=1}^{K} p_k + \sum_{k=1}^{K} u_k \,.$$

∎