[Reviews · NeurIPS 2015]

Submitted by Assigned_Reviewer_1

The authors investigate a class of combinatorial cascading bandit problems. The archetypal problem in this class seems to be that of routing packets in a network where edges are subject to random independent failures. I will restrict to this problem to simplify the description (though the authors also discuss the disjunctive version of this reward function). At each time step the agent chooses a path and receives unit reward only if none of the edges along the path failed. If some edge failed the agent gets a reward of zero and is also given the feedback of which was the first edge that failed. The algorithm they design for this problem called CombCascade, essentially builds UCB estimates for edge failure probability and picks the path with highest expected non-failure probability. They prove upper bounds for the regret of their algorithm one that is gap-dependent and one that is not. They argue that the upper bound matching the previously known tight upper bound for the stochastic semi-bandit problem (which is a more informative feedback setting) perhaps indicates that their algorithm is tight in settings that are "close" to the semi-bandit setting. The paper concludes with some experiments on synthetic and real datasets that show sub-linear growth of regret for their algorithm and they show that it performs better than CombUCB1 which is a stochastic combinatorial semi-bandit algorithm.

There are a couple of limitations of the study. One, as the authors also point out, is the rather strong independent failure assumption of the graph edges. [Aside: I appreciate the authors' honest comments pointing out the strongest and weakest parts of their paper, I wish more papers did that, not so much to simplify the referee's task, but to stimulate any reader of the paper.] Secondly, their work points to tantalizing connections to the semi-bandit setting, but this is not clarified well. This is perhaps food for future work. For instance, the weight estimation stage of their algorithm hints at a possible reduction between the two settings. On the other hand, the presence of interesting practical problems in their setting argues for acceptance of the paper despite these issues.
Summary: The authors design bandit algorithms in a new combinatorial cascading bandit setting. They prove regret bounds for the algorithm they design. They evaluate their algorithm on synthetic and real examples.

Submitted by Assigned_Reviewer_2

Summary: This paper formulates the combinatorial cascading bandit problem where the decision space consists of sets of items and reward is defined in a conjunctive or disjunctive manner. This problem is different from the well-known combinatorial semi-bandit setting whose reward is defined as a linear function of the decision. The authors propose an algorithm for the bandit problem and prove gap its dependent/independent O(log n) regret bounds (n: # of trials). Finally, the authors show experimental results for three real data sets

Comments: Quality: The technical contribution of the paper is presented very well in terms of both theoretical and empirical aspects. The regret bounds are reasonable, but there is no lower bound yet.

Time complexity of the proposed algorithm depends on the maximization problem of the objective f over the decision set A. It would be nice if the authors could discuss more details about how efficiently the optimization problem can be solved for particular combinatorial decision sets

Clarity: The paper is concisely organized and well-written.

Originality: An interesting contribution of the paper is to propose a bandit problem with a nonlinear objective (disjunctive or conjunctive) and its solution. This contribution seems new, whereas previous approaches consider linear objectives. However, curiously, the analysis of the paper heavily depends on the previous work of Kveton et al.

Significance:

The significance of the paper lies in the bandit algorithm for optimizing conjunctive/disjunctive and the analysis

Summary: This paper investigates a new combinatorial stochastic bandit problem with a nonlinear objective.The solution to optimize the nonlinear objective is significant.

Submitted by Assigned_Reviewer_3

The authors study a setting called combinatorial cascading bandits, where the utility is the minimum utility of a set of actions, and the feedback is the observed utility and the first action that causes the utility to be 0 (utility is either 1 or 0).

Algorithms are proposed with theoretical guarantees and empirical validation.

As a whole, this paper is solid, with rigorous problem definitions, algorithm develop & analysis, and empirical validation.

However, the research contribution of this paper compared to the CascadeUCB paper seems a bit marginal, and I'm having a hard time determining the magnitude of the contribution.

I hope the authors can more clearly state the technical novelty of this work compared to previous work.

Right now, it appears to just be a relatively modest twist, with a slightly modified algorithm and basically the same type of analysis.

So I'm not sure if the contributions justify a NIPS paper.

For instance, it's possible I'm under-estimating the technical challenges in adapting the theoretical analysis from previous work to this setting.

Minor comments:

-- The paper is missing a discussion of the relationship with ranked bandits, especially with the disjunctive objective version.

*** RESPONSE TO AUTHOR REBUTTAL *** I thank the authors for their rebuttal.

The rebuttal sharply describes the difference from previous work, and has suitably addressed my main concerns.
Summary: This paper is solid and well-written, but the magnitude of the research contribution relative to previous work is unclear.

Author Feedback
Author rebuttal: Dear reviewers,

We wanted to thank you for your reviews, praising the technical quality of our paper and its presentation, and suggesting acceptance in 4 out of 6 cases. Our rebuttal addresses two major issues raised by your reviews: novelty with respect to [9] and [11], and tightness of our analysis.

*** Novelty ***

Our work is novel in several aspects:

1) We propose a new learning setting for stochastic partial monitoring with combinatorial action sets that can model many interesting real-world problems. One of these problems is learning a routing path that fails with the lowest probability. The feedback is the index of the first failing path segment. Another problem is learning to recommend a list of items subject to diversity constraints that satisfy the user with the highest probability. The feedback is the index of the first satisfactory item. We evaluate our learning algorithm on two real-world datasets and show that it is practical.

2) Our algorithm, CombCascade, is not that of [9] and [11]. CombUCB1 [11] chooses solutions with the largest *sum* of the UCBs of items. CascadeUCB1 [9] chooses K items with the largest UCBs. CombCascade chooses solutions with the largest *product* of the UCBs of items. In the setting of [9], any list of K items is feasible, both CascadeUCB1 and CombCascade can learn the optimal solution. However, under general constraints, it is critical to maximize the product of the UCBs. CombUCB1 may lead to suboptimal solutions and we show this in Section 4.1.

3) The proof of Theorem 1 is completely different from those of Theorems 2 and 3 in [9]. The proofs in [9] are based on counting the number of times that each suboptimal item is chosen instead of any optimal item. This argument requires that the items in the solutions are exchangeable. Therefore, it can be only applied to special combinatorial sets, such as matroids. We utilize a part of the analysis of [11] (pages 12 and 13) to achieve linear dependency on K in Theorem 1. The rest of our analysis, including all technical lemmas and the proof of the gap-free bound, is completely novel.

Our main technical contribution is the reduction of our analysis to that of stochastic combinatorial semi-bandits. Roughly speaking, the analysis of CombCascade can be trivially reduced to semi-bandits by conditioning on the event of observing all items (lines 231-237), and increasing the expected regret for choosing any solution A from \Delta_A to \Delta_A / p_A, where p_A is the probability of observing all items in A. This reduction is problematic because it overestimates the regret when p_A is small and yields a regret bound that depends on a potentially huge \max_A (1 / p_A).

We address this issue by formalizing the following insight into our problem. If the expected reward for choosing A is small, the learning agent can distinguish A from A^\ast without knowing the expected weights of all items in A (lines 238-243). Technically speaking, the learning agent acts implicitly on the prefixes of suboptimal solutions, and we choose them in our analysis such that the probability of observing all items in these prefixes is "similar" to that of observing all items in A^\ast, and the gaps are "similar" to those of the original solutions. Then we count the number of times that the prefixes are chosen instead of A^\ast when all items in the prefixes are observed. Our analysis relies heavily on several structures in our problem, such as that our reward and observation models are closely related, [expected reward for choosing A] = p_A x [expected weight of the last item in A]. We are unaware of any prior work that exploits such structures in combinatorial learning. Our reduction yields an extra factor of 8 / [expected reward for choosing A^\ast], which is typically a significant improvement over \max_A (1 / p_A).

*** Tightness of our analysis ***

The lower bound in Theorem 4 [9] applies to the disjunctive variant of our problem and is \Omega((L - K) (1 / [K \Delta]) \log n) for p = 1 / K. This indicates that our upper bound in Theorem 1 is off by a factor of at most O(K^2). We believe that this factor is intrinsic to CombCascade, and we show this on the following class of problems. The ground set are L = K^2 items. The feasible set are K tuples:

\Theta = {(1, ..., K), (K + 1, ..., 2 K), ..., (L - K + 1, ..., L)}.

The expected weights of the first K items are 1. The expected weights of the remaining L - K items are (1 - K \sigma)^{1 / K}. Therefore, the expected rewards for the optimal and suboptimal solutions are 1 and 1 - K \sigma, respectively. The gap is K \sigma. Our empirical results are reported at:

https://dl.dropboxusercontent.com/u/43441773/RegretCombCascade.pdf

For all \sigma, the regret in n = 10^6 steps is linear in L - K. This is suggested by our upper bound, which is O(K (L - K) (1 / [K \sigma]) \log n). We do not know yet if the gap between the upper and lower bounds can be closed.